# Novel Function of Nogo-A as Negative Regulator of Endothelial Progenitor Cell Angiogenic Activity: Impact in Oxygen-Induced Retinopathy

**DOI:** 10.3390/ijms241713185

**Published:** 2023-08-24

**Authors:** Pakiza Ruknudin, Ali Riza Nazari, Maelle Wirth, Isabelle Lahaie, Emmanuel Bajon, Alain Rivard, Sylvain Chemtob, Michel Desjarlais

**Affiliations:** 1Department of Ophthalmology, Maisonneuve-Rosemont Hospital Research Center, University of Montréal, Montréal, QC H1T 2H2, Canada; 2Departments of Pediatrics, Ophthalmology and Pharmacology, Centre Hospitalier Universitaire Sainte-Justine Research Center, Montréal, QC H1T 2H2, Canada; 3Department of Medicine, Centre Hospitalier de l’Université de Montréal (CHUM) Research Center, Montréal, QC H1T 2H2, Canada

**Keywords:** Endothelial Progenitor Cell (EPC), angiogenesis, revascularization (RV), oxygen-induced retinopathy (OIR), Nogo-A, NgR1

## Abstract

Endothelial Progenitor Cells (EPCs) can actively participate in revascularization in oxygen-induced retinopathy (OIR). Yet the mechanisms responsible for their dysfunction is unclear. Nogo-A, whose function is traditionally related to the inhibition of neurite function in the central nervous system, has recently been documented to display anti-angiogenic pro-repellent properties. Based on the significant impact of EPCs in retinal vascularization, we surmised that Nogo-A affects EPC function, and proceeded to investigate the role of Nogo-A on EPC function in OIR. The expression of Nogo-A and its specific receptor NgR1 was significantly increased in isolated EPCs exposed to hyperoxia, as well as in EPCs isolated from rats subjected to OIR compared with respective controls (EPCs exposed to normoxia). EPCs exposed to hyperoxia displayed reduced migratory and tubulogenic activity, associated with the suppressed expression of prominent EPC-recruitment factors SDF-1/CXCR4. The inhibition of Nogo-A (using a Nogo-66 neutralizing antagonist peptide) or siRNA-NGR1 in hyperoxia-exposed EPCs restored SDF-1/CXCR4 expression and, in turn, rescued the curtailed neovascular functions of EPCs in hyperoxia. The in vivo intraperitoneal injection of engineered EPCs (Nogo-A-inhibited or NgR1-suppressed) in OIR rats at P5 (prior to exposure to hyperoxia) prevented retinal and choroidal vaso-obliteration upon localization adjacent to vasculature; coherently, the inhibition of Nogo-A/NgR1 in EPCs enhanced the expression of key angiogenic factors VEGF, SDF-1, PDGF, and EPO in retina; CXCR4 knock-down abrogated suppressed NgR1 pro-angiogenic effects. The findings revealed that hyperoxia-induced EPC malfunction is mediated to a significant extent by Nogo-A/NgR1 signaling via CXCR4 suppression; the inhibition of Nogo-A in EPCs restores specific angiogenic growth factors in retina and the ensuing vascularization of the retina in an OIR model.

## 1. Introduction

The retinopathy of prematurity (ROP) is the leading cause of visual impairment and blindness in preterm infants. It is characterized by an initial phase of vascular degeneration that will trigger a pathological retinal neovascularization (NV). One of the key factors associated with the development of ROP is the relative hyperoxia that premature infants are exposed to compared with the intra-uterine environment [1,2]. Hyperoxia is toxic to the immature retina of infants by triggering a pro-oxidant/inflammatory microenvironment that suppresses the production of key angiogenic growth factors such as VEGF, SDF-1, PDGF, and EPO, necessary for proper vascular development, resulting in degeneration of the microvasculature [1,2]. 

To date, focus has been placed on anti-angiogenic therapy to tackle pathologic pre-retinal NV, but efforts in promoting revascularization during the initial phase of the vascular degeneration of ROP and limiting subsequent intra-vitreal NV have been limited [3,4]. Pro-angiogenic therapies based on stem cell utilization have been documented for other ischemic cardiovascular pathologies such as hindlimb ischemia [5,6,7], myocardial infarct, and stroke [7,8], but reports on ischemic retinopathies are more limited, particularly as it applies to OIR/ROP. This inference especially applies to (bone marrow-derived) endothelial progenitor cells (EPCs), which are known to contribute to vascular development and endothelial repair via vasculogenic and paracrine actions [9]. EPCs have the ability to directly integrate the NV site in ischemic tissues to participate in the formation of new vessels, as reported in the promotion of NV, and post-ischemic revascularization in models of ischemia [4,9,10]; EPCs also secrete proangiogenic factors such as VEGF, EPO, SDF-1, and PDGF to elicit paracrine actions [9]. However, if exposed to oxidative stress and inflammation, EPC populations drop and their function is compromised [11]; the impact of the latter in ROP remains unknown. 

Repulsive anti-angiogenic factors contribute to sustaining vaso-obliteration [12] until inflammation is abated [13] and controlled [14]; in this process, Semaphorin 3A has been shown to participate in vaso-obliteration. However, its effects do not fully explain the delay in revascularization, inferring a role for other factors. Nogo-A, an important member of the reticulon protein family (RTN4), which includes Nogo-B and C, is traditionally associated with the development of the central nervous system [15], which acts as an anti-angiogenic and repulsive factor on endothelium [16] and participates in retinal vascularization, as seen in the OIR model of ROP [16,17]. Nogo-A disrupts the coordinated migration between endothelial tips and stalks cells [16]. Nogo are transmembrane proteins containing an extracellular segment of 66 amino acids called Nogo-66 [18]. Nogo-66 binds with high affinity to the Nogo receptor (NgR1) and activates the RhoA signaling pathway, resulting in a destabilization of the actin cytoskeleton as well as the collapse of the neurite growth cone [19]; a fragment of Nogo-66 (1–40) is known to inhibit Nogo-A activity [20]. The endothelial cell (EC)-repellent activity of Nogo-A is segment 20 (Nogo-20)-dependent [16], which binds to the sphingosine 1-phosphate receptor (S1PR2) to induce RhoA pathway activation [21]; conversely, suppression of S1PR2 accelerates the vascularization of the central retina [22]. However, the endothelial cell type involved in this process, specifically as it applies to relevant EPCs in ischemic retinopathy, remains unknown. Since the migration of EPCs are affected by their environment [23], we proceeded to explore the role of Nogo-A (and its receptor) in EPCs in the context of OIR. 

## 2. Results

### 2.1. Bone Marrow-Derived EPC and Angiogenic Property Validation

The phenotype of EPCs extracted from the bone marrow (BM) of rat pups at postnatal day 10 (P10) was based on an established culture method and approach (illustrated in Figure 1A). Total mononuclear cells were extracted from BM and cultured in an endothelial growth medium on a fibronectin matrix. After 21 days of culture, we validated the phenotype of adhered cells by evaluating their morphology, the presence of endothelial (lectin, CD31, or VEGFR2) and hematopoietic markers (CD34 or CD117), and their angiogenic activities. As shown in Figure 1B, the cells show EC morphology, positivity for the FITC-labeled lectin, and an ability to form tubules in Matrigel, consistent with EPC characteristics.

### 2.2. Inhibition of Nogo-A or NgR1 Signaling Rescues the Angiogenic Activity of EPCs Otherwise Impaired by Hyperoxia

The ex vivo exposure of EPCs to hyperoxia (80% O_2_) induced a time-dependent increase in Nogo-A and NgR1 expression (mRNA) for the first 24 h, followed by a return to baseline over the following 24 h (Figure 2A). A comparable surge in Nogo-A and NgR1 in EPCs from animals exposed in vivo to hyperoxia (from P5–P10) was also observed (Figure 2B).

To assess the co-regulation of other major pro- and anti-angiogenic factors in hyperoxia-exposed EPCs, we first studied their expression profile. Of these, the most affected were SDF-1, CXCR4, and HIF-1α, all of which were suppressed by hyperoxia (Figure 3A). In an attempt to determine if these factors were regulated by Nogo/NgR1, we used Nogo-A inhibitor (neutralizing peptide) and siRNA-NgR1 (Appendix A). Of all the factors studied, only SDF-1 and CXCR4 were rescued by the Nogo-A inhibitor and siRNA-NgR1 (Figure 3B), indicative of their specific regulation by Nogo-A. Concordantly, EPC migration and tubulogenesis were enhanced by the Nogo-A inhibitor and siRNA-NgR1 (Figure 4A,B); the effects of the Nogo-1 inhibitor and NgR1 silencing were annulled when CXCR4 was suppressed (by siRNA-CXCR4), consistent with the regulation of CXCR4 by Nogo-A. Likewise, choroidal ex vivo sprouting was affected by the Nogo-1 inhibitor and siRNA-NgR1 in the absence or presence of siRNA-CXCR4 (Appendix A).

### 2.3. Nogo-A-Inhibited EPCs Enhance Retinal Revascularization in OIR

Based on the in vitro results, we next determined if the systemic (intraperitoneal) administration of EPCs pre-treated with a Nogo-A neutralizating peptide or NgR1 siRNA diminishes vaso-obliteration. Nogo-A inhibitor and NgR1-suppressed EPCs attenuated retinal vaso-obliteration (and restored choroidal vascular thickness) in rats subjected to OIR, thus improving vascularization (Figure 5A and Appendix A); the concurrent suppression of CXCR4 abolished vascular benefits incurred by inhibiting Nogo-A, pointing to a prominent role for CXCR4 in mediating the vascular effects of Nogo-A. Interestingly, intraperitoneally administered EPCs (GFP-tagged) localized immediately adjacent to the retinal vasculature, particularly when Nogo-A/NgR1 was inhibited/suppressed (Figure 5B); this was consistent with an increase in EPC markers (CD34, CD117, CD133, CXCR4) in the retina (Figure 5C). Coincidentally, (at P10) the retinal expression of major pro-angiogenic factors VEGF, SDF-1, PDGF, and EPO was markedly augmented by Nogo-A-supressed or NGR1-supressed EPCs (Figure 5D).

## 3. Discussion

Neovascularization (NV) and post-ischemic revascularization involves the active participation of several cell types including bone marrow-derived EPCs. EPCs (CD34+/VEGFR2/CD133+) are further divided into pro-angiogenic-hematopoietic bone marrow-derived cells labelled PACs (early EPCs) that mostly act in a paracrine manner, and late outgrowth EPCs, also referred to as endothelial colony-forming unit cells (ECFC) [4,24,25,26]; the latter possess many features of mature endothelial cells, including the ability to form new vessels by themselves (vasculogenesis). PACs and ECFC complement each other in post-ischemic revascularization [4,24,25,26]. In contrast to PACs, the precise mechanisms regulating EPCs (ECFC) function in ischemic retinopathy, particularly in retinal vaso-obliteration associated with OIR, remains unexplored. However, EPCs often fail to exert the desired revascularization benefits in conditions otherwise requiring it, such as in stroke, peripheral ischemia, and myocardial infarct. A number of risk factors associated with these conditions, notably aging, dyslipidemia [27,28], hyperglycemia [29], and hypertension [30], confer a microenvironment characterized by oxidative stress intertwined with sustained systemic inflammation; these conditions compromise EPC numbers and function. This pro-oxidant/inflammatory microenvironment is also present in OIR [1,31], largely due to the hyperoxic environment. We therefore aimed to investigate the impact of hyperoxia on the functional activity of EPCs and explored potential targets that participate in this pathophysiological process. Based on the premise that repulsive factors could interfere with the revascularization of the retina, we focused on Nogo-A—an important member of the reticulon protein family (RTN4). Our findings identify an unprecedented significant role for Nogo-A, specifically in EPCs, in curtailing the revascularization of the retina subjected to OIR through the downregulation of SDF-1/CXCR4; the ensuing enhanced vascularization is equally observed in the choroid compromised in OIR [32]. A schematic diagram depicting the role of Nogo-A in EPCs on retinal vascularization is presented in Figure 6.

An important feature of this study is that Nogo-A is upregulated in EPCs subjected to hyperoxia (ex vivo and in vivo), and these cells exhibit a limited ability for the revascularization of the retina compared with EPCs engineered upon the suppression of Nogo-A. These findings are consistent with previous studies on EPC function in ischemic conditions such as hindlimb ischemia, stroke, and myocardial infarct [33,34]. The role of Nogo-A stands out in the present study, and speculatively for other ischemic conditions addressed. Nogo-A acts as a negative regulator of central nervous system angiogenesis, through its intracellular Nogo-A delta20 fragment domain [16,20]. MicroRNAs—important negative regulators of gene expression [35,36]—possibly modulate the expression of Nogo [37] during OIR [3]. For instance, miR-182-5p regulates Nogo-A expression [38] and promotes neurite outgrowth of hippocampal neurons in vitro; interestingly, this miR is decreased by hyperoxia, resulting in a rise in Nogo-A [3].

The inhibition of Nogo-A was found to increase the expression of a key mobilization factor, notably SDF-1, in EPCs. SDF-1 exerts an important role in vascular repair by inducing the EPC migration and proliferation largely ascribed to its action on CXCR4 [39], which is itself also regulated by Nogo-A, as observed herein. The control of SDF-1 by Nogo-A has been shown to occur through the downregulation of miR-141-3p and miR-454 [40,41]. Nogo-A can also modulate the angiogenesis activities of microvascular ECs via Ras/Rho-A and ROCK kinase activation [16], two signaling pathways that are involved in cellular proliferation and migration [42]. Nogo-A also affects the specific S1PR2 receptor to trigger the activation of RhoA/ROCK signaling [43]. ROCK is, in turn, the effector protein of the small GTPase RhoA [44], through MYPT-1, LIMK, and ERK1/2 pathways [45], to cause the vascular permeability, migration, and proliferation of ECs and the formation of capillaries [46], but, depending on the cell type, the RhoA pathway can either inhibit or promote angiogenesis [47,48,49].

Although we focused on and identified an important role for Nogo-A in EPCs in regulating retinal angiogenesis, other members of the Nogo family also partake in governing angiogenesis through distinct signaling. For instance, the Nogo-B receptor (NgRB) is essential for angiogenesis in zebrafish because it signals through the Akt pathway [50]. NgRB knockdown abolishes its specific ligand activity, as well as Nogo-B-stimulated endothelial cell migration, by reducing the VEGF-stimulated phosphorylation of Akt and downstream chemotaxis and the morphogenesis of human umbilical vein ECs; this suggests a pro-angiogenic role for Nogo-B. It should also be highlighted that Akt-triggered prosurvival and angiogenesis is eNOS-dependent, whereby NO assists in EPC recruitment in ischemic tissues [51,52]; hence, NO amplifies the recruitment of PAC/EPCs into vessels in neovascular sites. Although, in this project, we do not focus on vascular permeability, some evidence suggests that NOGO can also indirectly modulate vascular permeability. For instance, in Figure 5D, we observed an increased retinal level of both VEGF and SDF-1 in the group of rats treated with EPC reprogrammed with a NOGO inhibitor or also with siNGR1. VEGF is well known to increase endothelial permeability by activating PKB/Akt, endothelial nitric-oxide synthase, and MAP kinase pathways [53,54]. SDF-1 per se also partakes in enhancing retinal vascular permeability [55,56].

In summary, we hereby report that hyperoxia impairs the activities of EPCs through the upregulation of Nogo-A, which curtails cell migration and vasculogenesis, thus compromising retinal revascularization in OIR rats; the effects of Nogo-A are largely attributable to the suppression of SDF-1/CXCR4. This study reveals, for the first time, that Nogo-A-suppressed EPCs are more effective than native EPCs in attenuating the retinal vaso-obliteration of OIR rats. Targeting Nogo-A in EPCs improves their functional activities and provides a new strategy to restore vascular integrity by reducing vaso-obliteration through revascularization in ROP/OIR.

## 4. Materials and Methods

### 4.1. Animal Care

All animal experimental procedures were performed with strict adherence to the ARVO Statement for the Use of Animals in Ophthalmic and Vision Research and approved by the Animal Care Committee of the Hospital Maisonneuve-Rosemont in accordance with guidelines established by the Canadian Council on Animal Care.

### 4.2. Identification of Endothelial Progenitor Cells (EPCs)/Morphological Characteristics

Bone marrow-derived endothelial progenitor cells (BM-EPCs), also known as “late outgrowth EPCs” or “endothelial colony forming cells” (ECFC), express hematopoietic stem cell myeloid markers, such as CD34 and CD117, and endothelial markers, such as lectin, VEGFR2, and CD31 [24,57,58]. Also, EPCs acquire EC morphology and are capable of forming tubes in vitro. The vast majority of adherent cells were found to bind FITC-labeled lectin, had EC morphology, formed tubes, and migrated in vitro. Accordingly, as reported, the morphological and functional characteristics of these cells are consistent with EPCs [4,10,27].

### 4.3. EPCs Isolation from Rat Bone Marrow

Mononuclear cells were isolated from the femoral and tibial bone marrow of Sprague Dawley rats on postnatal day 10 (P10). Bone marrow was flushed with medium 200 (Thermos Fisher, Burlington, ON, Canada) supplemented with 10% fetal bovine serum (FBS, Wisent, St-Jean-Baptiste, QC, Canada), 100 IU/mL penicillin/0.1 mg/mL streptomycin (Wisent, St-Jean-Baptiste, QC, Canada), and low serum growth supplement (LSGS; 2% FBS, 3 ng/mL bFGF, 10 mg/mL heparin, 1 mg/mL hydrocortisone, and 10 ng/mL EGF; Thermos Fisher, Burlington, ON, Canada) to harvest the cells. The cells were kept 21 days in culture on fibronectin-precoated plates (MilliporeSigma, Oakville, ON, Canada) enabling EPC phenotype characteristics. After 21-days of culture, the adherent cells were photographed and stained (1 h) with FITC-labeled lectin BS-1 (Bandeiraea simplicifolia, Vector Laboratories, Newark, CA, USA). Matrigel tube formation assay was also performed. The adherent cells were plated at a density of 30,000 cells/well in 96-well plates precoated with 50 μL of growth factor reduced Matrigel Matrix (Fisher Scientific, Waltham, MA, USA) and cultured at 37 °C for 6 h in normoxia in complete endothelial growth medium. The cells showing EC phenotype, positivity for lectin, and tube formation were considered as EPCs [4,10,25,27].

### 4.4. Oxygen-Induced Retinopathy (OIR)/Vaso-Obliteration Model

The angiogenic properties of EPCs and reprogrammed EPCs were studied during the vaso-obliteration phase of OIR by engineering EPCs using either a NOGO inhibitor (Nogo-66-neutralized peptide) or a siRNA targeting NgR1 or CXCR4. OIR phenotype was generated by exposing Sprague Dawley newborn rats to a constant hyperoxia (80% O_2_) from P5 to P10 [4]. Gas delivery to chambers was controlled by a computer-assisted Oxycycler (BioSpherix, Parish, NY, USA). Three groups of rat pups were anesthetized 30 min before hyperoxia exposure at P5 and were intraperitoneally injected with 50 μL of 200,000 native EPCs or engineered EPCs (Nogo-66-neutralized-EPCs) or PBS used as a control. Some of the rat pups of each group were euthanatized at P8 and their retinas were collected for molecular analysis. The rest of them were euthanatized at P10 and their retinas were collected for vessel immunostaining (retinal flat mounts) and for molecular analysis; N = 4 retinas/group for qRT-PCR, 6 retinas/group for flat mounts, and 4 retinas/group for cross-section experiments.

### 4.5. Immunohistochemistry of Retinal Vessels

In order to study the retinal vasculature, enucleated eyes were fixed for one hour in 4% paraformaldehyde at room temperature and kept in PBS until retinal flat mount preparation. The retinas were incubated overnight in 1% Triton X100 and 1 mM CaCl2/PBS with the tetramethylrhodamine isothiocyanate–conjugated lectin endothelial cell marker Bandeiraea simplicifolia (1:100; Vector Laboratories, Newark, CA, USA). Before preparing microscope slides, PBS was used to wash retinas. They were mounted on microscope slides (Bio Nuclear Diagnostics, Inc., Toronto, ON, Canada) under coverslips with mounting media (Fluoro-Gel; Electron Microscopy Sciences, Hatfield, PA, USA). Then, an epifluorescence microscope was used to take pictures of retinas (Zeiss AxioObserver; Carl Zeiss Canada, Toronto, ON, Canada) and the MosiaX option in the AxioVision 4.6.5 software (Carl Zeiss Canada, Toronto, ON, Canada) was used to merge the images into a single file. For choroidal vasculature, retinal cross-sections were performed. Eyes were collected, dehydrated by alcohol, and embedded in paraffin. Sagittal sections (7 µm thick) were cut by microtome (RM 2145; Leica, Wetzlar, Germany). Posterior eyecups were frozen in an optimal cutting temperature medium and stained for choroidal vessels with TRITC-conjugated tetramethylrhodamine isothiocyanate-labeled lectin (MilliporeSigma, Oakville, ON, Canada) in the cryosections. Sections were then visualized with an epifluorescence microscope (Eclipse E800; Nikon, Tokyo, Japan). In the experiment, retinal cryosections or flatmount were co-stained (lectin/CD34) or GPF-labeled EPCs, by adding a rabbit antibody anti-CD34 (1:200, ab185732; ABCAM, Waltham, MA, USA), or GFP and incubated overnight at 4 °C in the blocking solution for CD34 staining. Secondary antibodies, such as Alexa Fluor 488 anti-rabbit (Thermos Fisher, Burlington, ON, Canada), were used at a dilution of 1:1000 to detect CD34.

### 4.6. Migration Wound Healing Assay

Cell migration and motility were analyzed using a scratch wound assay on confluent EPCs. The cells were grown to near confluence in 24-well plates and were treated or not with different doses of Nogo-66 antagonist peptide (MilliporeSigma, Oakville, ON, Canada) at different concentrations (1, 5, 10 μM) or treated with 50 μM of a siRNA control, siRNA-NGR1, or siRNA-CXCR4 (Thermos Fisher, Burlington, ON, Canada) for 24 h. After treating the cells, the monolayer scratch was performed mechanically using a pipette tip and cells were then exposed or not to hyperoxia (80% O_2_). Cell migration was evaluated 24–48 h later using an inverted microscope at a magnification of 100×. Cell density was quantified. Five fields per well were evaluated and all experiments were performed in duplicate.

### 4.7. EPC Capillary-like Tubulogenesis on Matrigel

The capacity of EPC to form capillary-like tubes was evaluated using a Matrigel assay. EPCs where treated or not with Nogo-66 antagonist peptide (10 μM), or treated with 50 μM of a siRNA control, siRNA-NGR1, or siRNA-CXCR4 (Thermos Fisher, Burlington, ON, Canada) for 24 h and plated at a density of 30,000 cells/well in 96-well plates precoated with 50 μL of growth factor reduced Matrigel Matrix (Fisher Scientific, Waltham, MA, USA) and cultured at 37 °C for 6 h in normoxia or hyperoxia in complete endothelial growth medium (see EPCs’ isolation from rat bone marrow section). A light microscope at a magnification of 10× was used to take pictures of the capillary-like tubes; these were quantified by counting branches and branching points.

### 4.8. Ex Vivo Choroidal Angiogenic Sprouting Assay

Angiogenic sprouting capacity of the choroid isolated from rats were assessed as previously described [59,60,61]. Briefly, choroid was isolated from rat pups at P10, sectioned into 1-mm rings, and placed into growth factor-reduced Matrigel (Fisher Scientific, Waltham, MA, USA) in 24-well plates and cultured in hyperoxia (80% O_2_) for 5 days in endothelial growth medium (EGM), used as positive control medium, as previously described, or treated in hyperoxia with Nogo-66 antagonist peptide (10 μM), 50 μM of a siRNA control, siRNA-NGR1, or siRNA-CXCR4 (Thermos Fisher, Burlington, ON, Canada), for 24 h. Photomicrographs of individual explants were taken at day 5 using an inverted phase-contrast microscope (AxioObserver; Carl Zeiss Canada, Toronto, ON, Canada), and microvascular sprouting (total area occupied by vessel sprouts excluding the explant) was quantified using Image J (ImageJ2).

### 4.9. qRT-PCR Analyses

In order to quantify the mRNA levels of Nogo-A, NgR1, and other angiogenic factors in EPCs and in rat retinas, RNeasy mini kit (Qiagen, Toronto, ON, Canada) was used to extract total RNA using the manufacturer’s protocol. To generate cDNA, iScript-II RT kit (Qiagen, Toronto, ON, Canada) was used for reverse transcription as per manufacturer’s protocol. In total, 25 ng of cDNA sample, 2 μM of specific primers (Alpha DNA, Montreal, QC, Canada) for the selected mRNAs, and Universal SYBR Green Supermix (BioRad Mississauga, ON, Canada) were used for quantitative real-time PCR reaction. The instrument detection system, ABI Prism 7500 (Applied Biosystems, Foster City, CA, USA), allowed to calculate the relative expression (RQ = 2^−ΔΔ^CT) and normalize it to b-Actin and GAPDH.

### 4.10. Statistical Analysis

All the results are presented as mean ± SEM. Statistical significance was evaluated by a one or two-way ANOVA followed by a Bonferroni post hoc test. A value of *p* < 0.05 was interpreted to denote statistical significance.

## Figures and Tables

**Figure 1 ijms-24-13185-f001:**
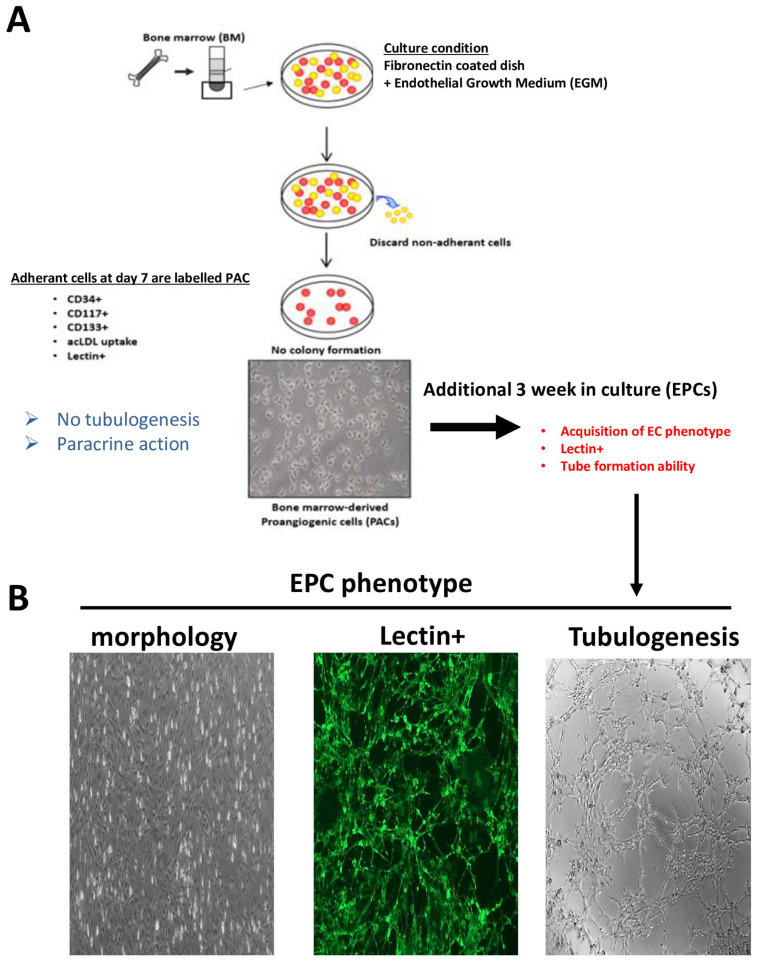
Phenotype validation of BM-isolated EPC. (**A**) Illustration of isolation of bone marrow-derived EPCs. Total mononuclear cells extracted from bone marrow were coated on a fibronectin matrix in EGM medium, and after 4 weeks in culture, the adhered cells were characterized and labelled to be EPCs. (**B**) EPCs display an endothelial-cell morphology (**left panel**) and are positive for FITC-labeled lectin BS-1 (**center panel**). EPCs have the ability of vasculogenesis in vitro in Matrigel assay (**right panel**).

**Figure 2 ijms-24-13185-f002:**
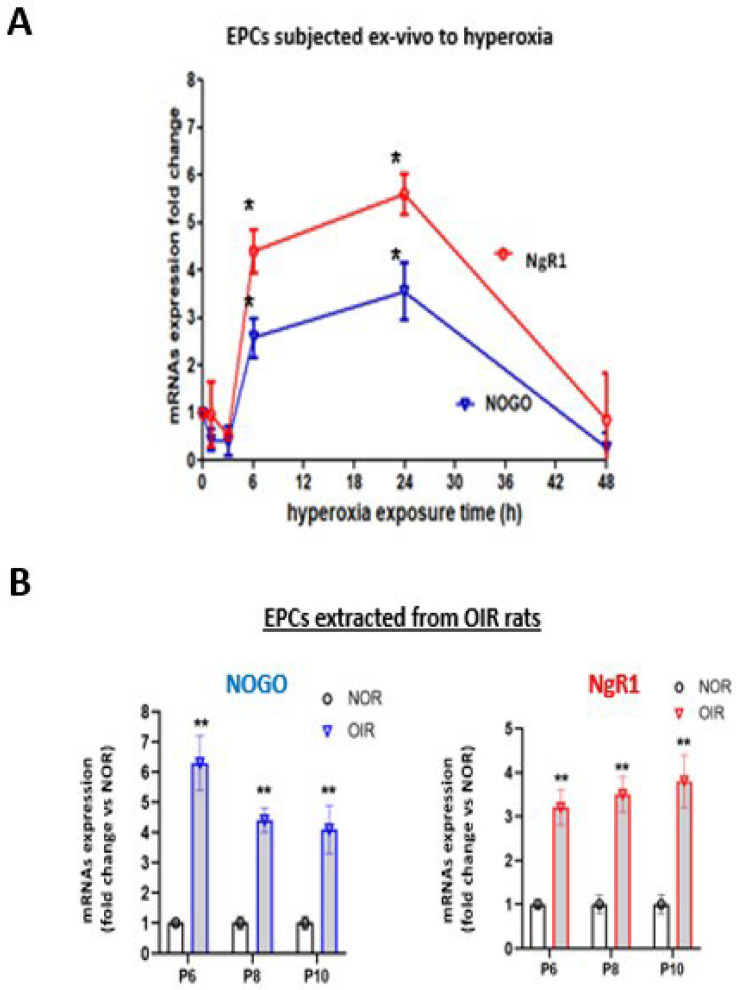
Nogo and NgR1 expression profile in OIR conditions. (**A**) mRNA expression levels of Nogo and its receptor NgR1 in EPCs subjected to hyperoxia (80% oxygen) at different time points (0 h, 2 h, 3 h, 6 h, 24 h, and 48 h). (**B**) NOGO and NgR1 mRNA expression in EPCs extracted from OIR rats at P6, P8, and P10. Data are mean ± SEM. * *p* < 0.05 vs. normoxia (control), ** *p* < 0.005 vs. normoxia (control), N = 3–4 experiments.

**Figure 3 ijms-24-13185-f003:**
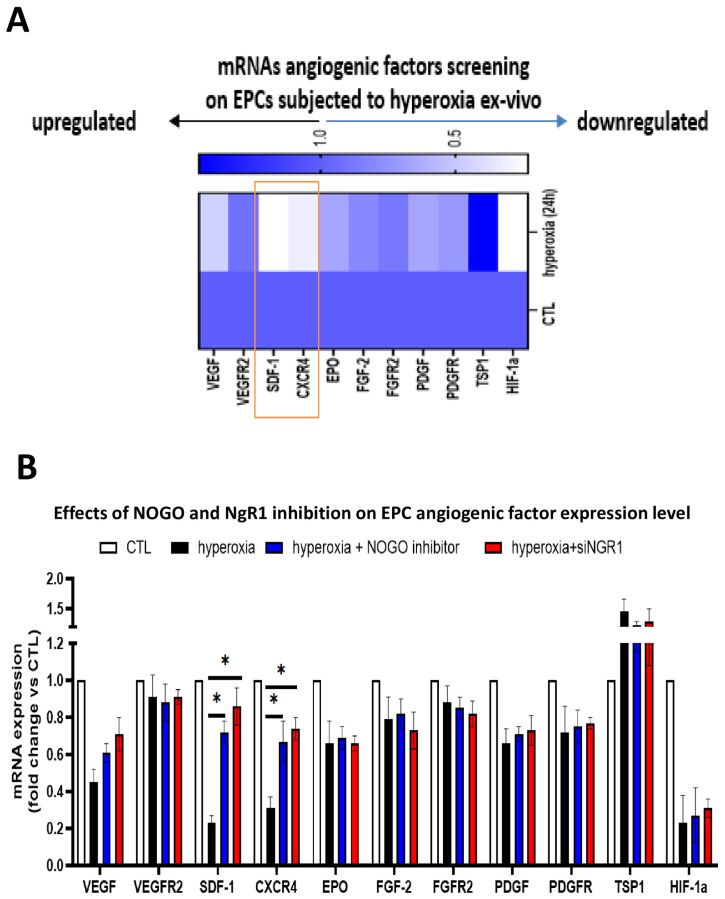
Inhibition of NOGO/NgR1 signalling restores SDF-1/CXCR4 in hyperoxic conditions. (**A**) qRT-PCR screening of major angiogenic factor expression in EPCs subjected to hyperoxia for 24 h. (**B**) Effects of NOGO inhibitor or siRNA-NgR1 in EPCs subjected or not to hyperoxia for 24 h, on growth factor mRNA expression. Data are mean ± SEM. * *p* < 0.05 vs. normoxia (control) or hyperoxia. N = 3–4 experiments.

**Figure 4 ijms-24-13185-f004:**
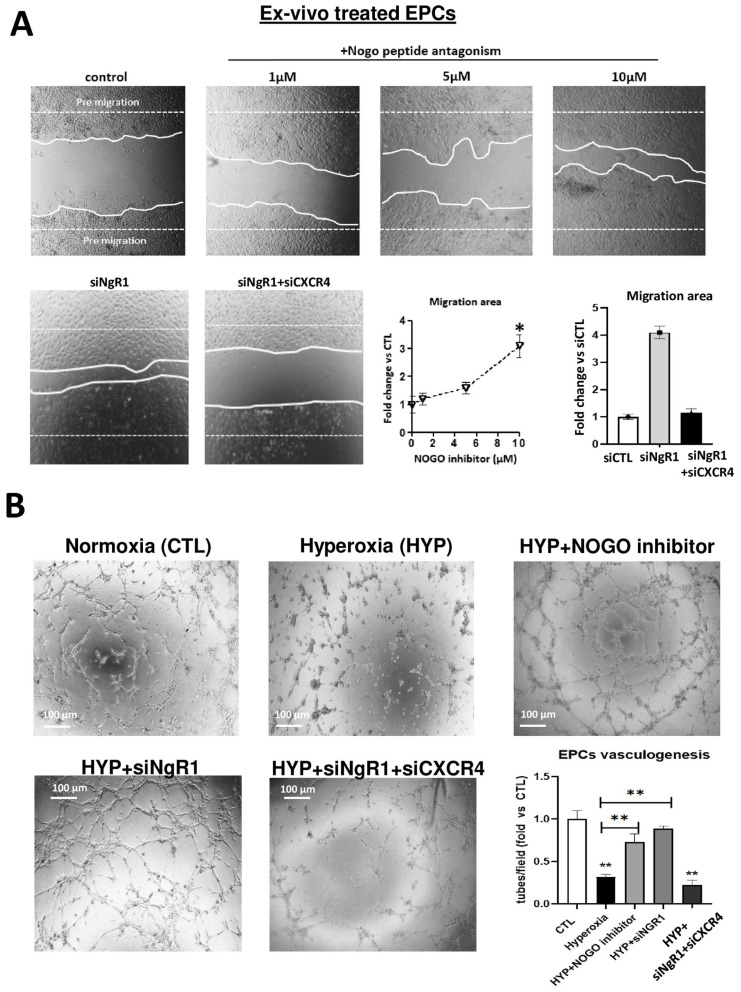
NOGO/NgR1 suppression reverse hyperoxia-induced dysfunction in EPC migration and tubulogenesis. (**A**) EPCs’ migration capacity after 24 h of treatment with NOGO inhibitor, siNgR1, or siNgR1 + siCXCR4 using a scratch migratory assay. (**B**) Tubule formation (Matrigel assay) of EPCs pre-treated or not with NOGO inhibitor, siRNA-NGR1, or siNgR1 + siCXCR4 subjected or not to hyperoxia; representative images (taken at 6 h) are shown for tubulogenesis. Compiled data are presented in linear and histogram format. Data are mean ± SEM. * *p* < 0.05 or ** *p* < 0.005 vs. normoxia (control) or hyperoxia. N = 4–5 experiments.

**Figure 5 ijms-24-13185-f005:**
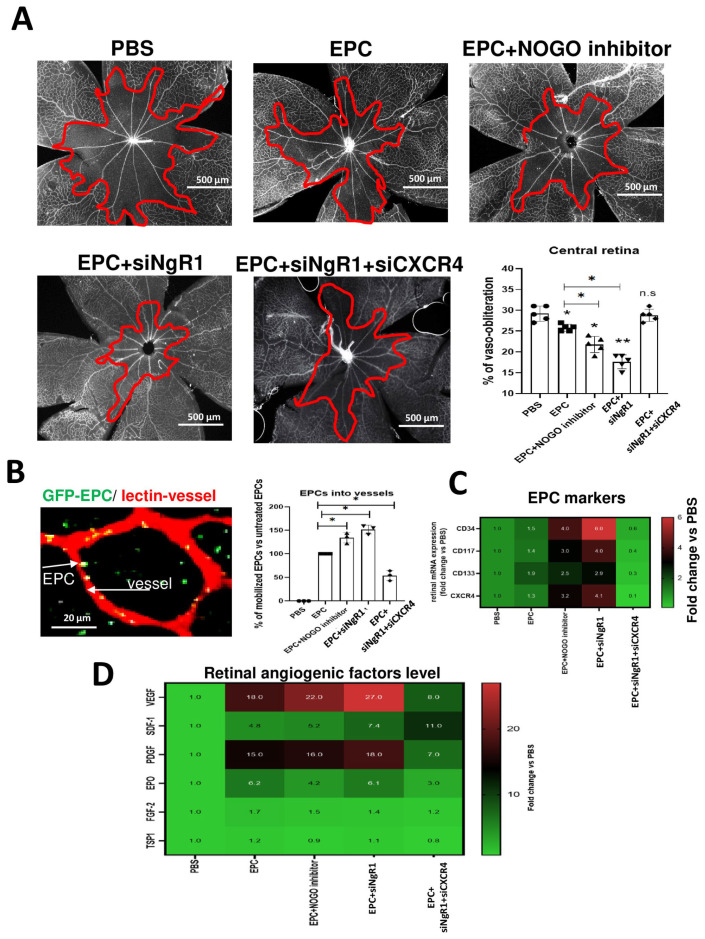
Effects of systemically administered EPCs engineered by inhibiting NOGO or suppressing NgR1, on vaso-obliteration in the OIR model in rats. (**A**) Representative flat mount photomicrographs and quantitative analysis of central retinal vaso-obliteration in OIR rats treated at P5 or not, with native, Nogo-inhibited EPCs, siNGR1-suppressed EPCs, or combined siNgR1 + siCXCR4-treated EPCs. (**B**) Representative image of GFP-labeled-EPCs mobilizing adjacent to retinal vessels after intra-peritoneal EPC injection; quantification analysis of number of EPCs localized at the vasculature in the different groups or rats (histogram). (**C**,**D**) Heat map showing qRT-PCR analyses, respectively, of EPC markers and angiogenic factor expression in the retina of OIR rats in the different groups of treated rats at P10. Data were mean ± SEM. * *p* < 0.05 or ** *p* < 0.005 vs. PBS (control). N = 4 retinas/group for qRT-PCR, 6 retinas for flat mounts, and 4 retinas for cross-section. Different shapes (circle, square, triangle, rhombus) are individual value of each animal (N).

**Figure 6 ijms-24-13185-f006:**
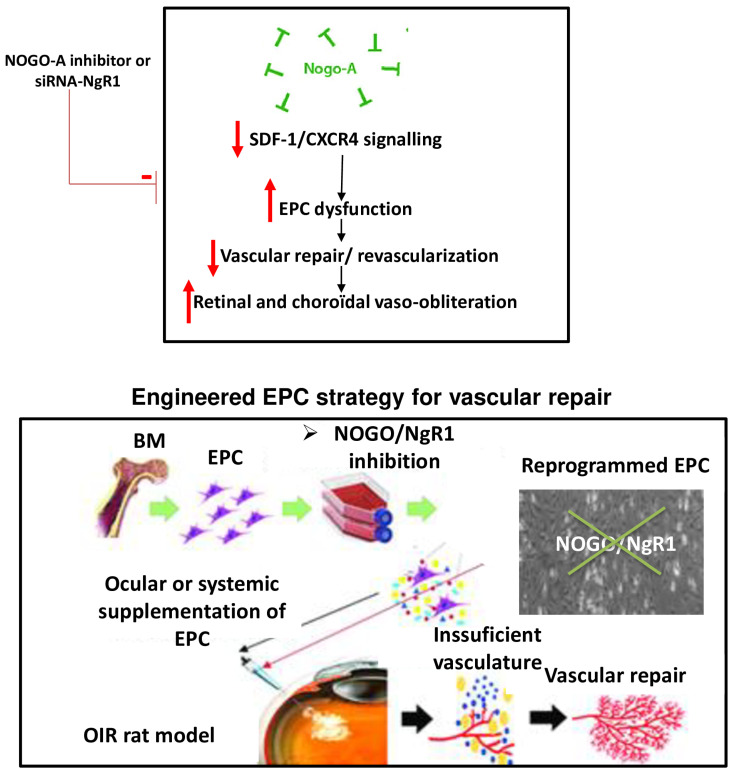
Schematic diagram illustrating the impact of engineered EPCs (inhibiting Nogo-A) on the ability of the vascular network to repair. A. The hyperoxic and OIR condition promotes the expression of Nogo-A leading to the activation of NgR1 that in turn downregulate SDF-1/CXCR4 signalling in EPCs leading to their vasculogenic dysfunction and increasing retinal vaso-obliteration. B. Systemic administration of engineered EPCs by suppressing Nogo-A activity can be a new therapeutic strategy for vascular repair in OIR.

## Data Availability

The data presented in this study are available on request from the corresponding authors.

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
