# Peer review of "Novel Function of Nogo-A as Negative Regulator of Endothelial Progenitor Cell Angiogenic Activity: Impact in Oxygen-Induced Retinopathy"

_ijms, 2023, doi:10.3390/ijms241713185_

Round 1
Reviewer 1 Report
This manuscript investigates the role of Nogo-A on bone-marrow SC-derived endothelial progenitor cells (EPCs) and in oxygen-induced retinopathy. Nogo-A is mainly known for its role in CNS development and vascularization. Nogo-A is upregulated when EPCs are exposed to hyperoxia, and this results in reduced revascularization. The results show that Nogo-A suppresses SDF-1 and CxCR4 expression, and that Nogo-A inhibitors (peptide or si-RNA) can prevent the effect of Nogo-A both in vitro (on EPCs isolated from bone marrow; after reaching confluence, scratch test of 1-2d cell migration with or without elevated oxygen; tubulogenesis assay on matrigel), ex vivo (choroidal angiogenic sprouting assay) and in vivo (in a rat model of oxygen-induced retinopathy: elevated oxygen P5-10, analysis of retinal wholemounts on P10). Experiments were analyzed by immunohistochemistry, qPCR and Western Blots.
General comments:
This is a well written paper with important results. The reviewer has only a few comments.
Specific comments:
p. 11, section 4.4: OIR model: The in vivo experiments consisted in exposing rat pups to 80% oxygen from postnatal day (P) 5 to P10. Rat pups were injected before onset of hyperoxia with 200K native EPCs or Nogo-66-neutralized EPCs or PBS. How many rats per experimental group? This should be listed here.
p. 11, line 272: “Gaz delivery” should be “Gas delivery”.
Figures: The figures are well designed. However, the reviewer is not sure whether the figures have the required resolution (may have been reduced too much when creating pdf). Please make sure that the final resolution is at least 300 dpi.
Quality of English language is fine, except for a typo (see comments above)
Author Response
Reviewer 1 : responses
General comments:
This is a well written paper with important results. The reviewer has only a few comments.
We thank the reviewer for the general appreciation of this paper.
Specific comments:
p11, section 4.4: OIR model: The in vivo experiments consisted in exposing rat pups to 80% oxygen from postnatal day (P) 5 to P10. Rat pups were injected before onset of hyperoxia with 200K native EPCs or Nogo-66-neutralized EPCs or PBS. How many rats per experimental group? This should be listed here.
The number of rats used in section 4.4 line 277, have been indicated; N=4 retinas/group for qRT-PCR, and 6 retinas/group for flat mounts and 4 retinas/group for cross-section experiments.
p11, line 272: “Gaz delivery” should be “Gas delivery”.
Typographical error for ‘Gas’ has been corrected.
Figures: The figures are well designed. However, the reviewer is not sure whether the figures have the required resolution (may have been reduced too much when creating pdf). Please make sure that the final resolution is at least 300 dpi.
Figure resolution is now above 300 dpi.

Reviewer 2 Report
The authors of the article “Novel function of Nogo-A as negative regulator of endothelial progenitor cell angiogenic activity: impact in oxygen-induced retinopathy” has demonstrated the effect of NOGO-A expression on the EPC malfunction.
The article is well written and explained.
Here are some questions for authors:
1. In figure 2B, what is the protein expression pattern of both the genes NOGO and NgR1 after 48 hrs?
2. Scale bars are missing in all the respective images.
3. In the material and methods explain the synthesis of engineered EPCs?
4. What is the effect of NOGO inhibitor on the vascular permeability? In order to check that the blood vessels recued after the NOGO inhibitor are leaky or more permeable.

Author Response
Comments and Suggestions for Authors
The authors of the article “Novel function of Nogo-A as negative regulator of endothelial progenitor cell angiogenic activity: impact in oxygen-induced retinopathy” has demonstrated the effect of NOGO-A expression on the EPC malfunction.
The article is well written and explained.
We thank the reviewer for appreciating our paper.
Here are some questions for authors:
- In figure 2B, what is the protein expression pattern of both the genes NOGO and NgR1 after 48 hrs?
In confirmatory experiments we did not observe any change in protein expression level for NOGO and NGR1 at 48 and 72 h (no upregulation), consistent with changes in mRNA.
- Scale bars are missing in all the respective images.
We have now added the scale bar in all images in the figures (fig4b and fig5a-b).
- In the material and methods explain the synthesis of engineered EPCs?
We thank the reviewer for pointing out this clarification. We have now specified the following in the method section 4.4 line 268: The angiogenic properties of EPCs were studied during the vasoobliteration phase of OIR by reprogramming EPCs using either a NOGO inhibitor (Nogo-66-neutralized peptide) or siRNA targeting NGR1 or CXCR4.
- What is the effect of NOGO inhibitor on the vascular permeability? In order to check that the blood vessels rescued after the NOGO inhibitor are leaky or more permeable.
This is an interesting comment. Although in this project we do not focus on vascular permeability some evidence suggests that NOGO can also indirectly modulate vascular permeability. For instance in figure 5D we observed an increased retinal level of both VEGF and SDF-1 in the group of rats treated with EPC reprogrammed with NOGO inhibitor or also with siNGR1. VEGF is well known to increase endothelial permeability by activating PKB/Akt, endothelial nitric-oxide synthase, and MAP kinase pathways (LAL et al, 2001; Venkatraman et al, 2019). SDF-1 alone is also sufficient to promote retinal vascular permeability (Butler et al, 2005 and Salvucci et al, 2004). These comments have been added to the Discussion; line 227.

Round 2
Reviewer 2 Report
The authors of the article “Novel function of Nogo-A as negative regulator of endothelial progenitor cell angiogenic activity: impact in oxygen-induced retinopathy” has well explained the questions raised.
The article can be considered for publication.
